# Role of ciliopathy protein TMEM107 in eye development: insights from a mouse model and retinal organoid

Marija Dubaic[1,2], Lucie Peskova[3], Marek Hampl[1,2], Kamila Weissova[1,3], Canan Celiker[3], Natalia A Shylo[4,5], Eva Hruba[1], Michaela Kavkova[6], Tomas Zikmund[6], Scott D Weatherbee[4,7], Jozef Kaiser[6], Tomas Barta[1,3], Marcela Buchtova[1,2]

Primary cilia are cellular surface projections enriched in receptors and signaling molecules, acting as signaling hubs that respond to stimuli. Malfunctions in primary cilia have been linked to human diseases, including retinopathies and ocular defects. Here, we focus on TMEM107, a protein localized to the transition zone of primary cilia. TMEM107 mutations were found in patients with Joubert and Meckel–Gruber syndromes. A mouse model lacking *Tmem107* exhibited eye defects such as anophthalmia and microphthalmia, affecting retina differentiation. *Tmem107* expression during prenatal mouse development correlated with phenotype occurrence, with enhanced expression in differentiating retina and optic stalk. TMEM107 deficiency in retinal organoids resulted in the loss of primary cilia, down-regulation of retina-specific genes, and cyst formation. Knocking out TMEM107 in human ARPE-19 cells prevented primary cilia formation and impaired response to *Smoothened* agonist treatment because of ectopic activation of the SHH pathway. Our data suggest TMEM107 plays a crucial role in early vertebrate eye development and ciliogenesis in the differentiating retina.

## Introduction

The primary cilium is a cellular organelle that projects from the surface of most cell types. It performs sensory, mechanical, and signal processing functions, and regulates numerous critical developmental processes, including neurogenesis, skeletogenesis, and kidney formation (Chang et al, 2015; Marra et al, 2016). Primary cilia are also crucial coordinators of the Sonic Hedgehog (Shh) pathway and other signaling processes (Huangfu et al, 2003; Caspary et al, 2007; Wheway et al, 2018). During early eye development, primary cilia are present on the surface of the optic neurepithelium, surface ectoderm, and periocular mesenchyme (Lupu et al, 2018). They also play an important role in the transport of proteins involved in visual transduction in photoreceptors, making ciliary signaling critical for both retinal structure formation and retinal physiology in postnatal stages (Wheway et al, 2014).

A broad range of defects known as ciliopathies can result from alterations in ciliary biogenesis or function (Lee & Gleeson, 2011). These defects include ocular deformities, such as retinitis pigmentosa and macular degeneration in humans, and eye defects observed in several mouse strains with ciliopathies (Badano et al, 2006; Gorivodsky et al, 2009; Qin et al, 2011; Cela et al, 2018). The protein composition and function of cilia is precisely controlled through a specialized domain located at the base of the cilium known as the transition zone (TZ) (Gonçalves & Pelletier, 2017). TMEM107 is a protein located at the TZ that has been shown to recruit ciliopathy-associated proteins such as MKS-1, TMEM-231 (JBTS20), and JBTS-14 (TMEM237) to this domain (Lambacher et al, 2016). The *TMEM107* locus has recently been found to be mutated in patients with Meckel–Gruber syndrome, Orofaciodigital syndrome, and Joubert syndrome (Iglesias et al, 2014; Shaheen et al, 2015; Lambacher et al, 2016; Shylo et al, 2016; Chinen et al, 2022). These syndromes are associated with altered primary cilia morphology and function, highlighting the role of TMEM107 in ciliary function.

Human patients with *TMEM107* mutations exhibit numerous developmental defects, such as polydactyly or facial dysmorphic features (Iglesias et al, 2014; Lambacher et al, 2016; Shylo et al, 2016; Chinen et al, 2022). Mouse strains with mutations in *Tmem107* demonstrate similar defects to human patients, including extra digits and a spectrum of craniofacial anomalies such as exencephaly, microphthalmia or skeletal defects in *Tmem107^schlei* embryos (Christopher et al, 2012). *Tmem107^null* mouse embryos display even stronger phenotypes, including shorter snouts, expanded facial midlines, cleft palates, and extensive exencephaly (Cela et al, 2018). Whereas the association between *Tmem107* deficiency and craniofacial defects has been previously described, the role of this gene in eye development is not yet fully understood.

Here, we used mouse embryos, retinal organoids, and retinal cell culture models to closely investigate the role of TMEM107 in eye

[1]Laboratory of Molecular Morphogenesis, Institute of Animal Physiology and Genetics, Czech Academy of Sciences, Brno, Czech Republic   [2]Department of Experimental Biology, Faculty of Science, Masaryk University, Brno, Czech Republic   [3]Department of Histology and Embryology, Faculty of Medicine, Masaryk University, Brno, Czech Republic   [4]Department of Genetics, Yale University, School of Medicine, New Haven, CT, USA   [5]Stowers Institute for Medical Research, Kansas City, MO, USA   [6]CEITEC - Central European Institute of Technology, Brno University of Technology, Brno, Czech Republic   [7]Biology Department, Fairfield University, Fairfield, CT, USA

Correspondence: buchtova@iach.cz; tbarta@med.muni.cz

development. We found that *Tmem107* is specifically enriched in the neural retina (NR) during mouse eye development. Its loss leads to the distinctive ocular phenotypes associated with primary cilia defects, including microphthalmia and anophthalmia; and altered expression of crucial transcription factors involved in eye development. *TMEM107*-deficient human retinal organoid model enabled us to determine its role in the human retina, and it also allowed us to study the role of this gene in neural retina formation without any influence of surrounding or closely associated eye structures and tissues including the brain or surface ectoderm. We found that *TMEM107*-deficient retinal organoids largely corroborated the results from the mouse model, failing to form neural retina structures and exhibiting primary cilia defects. Finally, using the retinal cell culture model, we found that *TMEM107* is critical for SHH signaling and its loss aberrantly up-regulates the SHH pathway. Taken together, our findings suggest that TMEM107 plays a crucial role in eye development and SHH signaling in both mice and humans, providing a better understanding of eye abnormalities that may potentially lead to therapeutic interventions for related conditions.

# Results

### Loss of *Tmem107* leads to distinctive ocular phenotypes in mouse embryos

We used $Tmem107^{-/-}$ mouse model to determine the function of *Tmem107* in eye development (Christopher et al, 2012). Mutant embryos through all analyzed stages (E10.5–E15.5) exhibited a variety of eye abnormalities (Figs 1A and S1A–F and Table S1) (Video 1 and Video 2). Heterozygous embryos did not exhibit an abnormal phenotype. Two most frequently observed phenotypes included complete loss of an eye (*anophthalmia*), observed in 33.3% of examined mutants, and abnormally small eye (*microphthalmia*) with 47% occurrence (Fig 1D). In nine examined cases, both phenotypes were present within the same embryo. In anophthalmic animals, a small area of pigment residue was present, whereas embryos exhibiting microphthalmia often displayed other defects, such as total absence of the lens (*aphakia*) and/or optic nerve (ON) hypoplasia (Fig 1A). The mice retina with microphthalmic phenotype was smaller and elongated, compared with retina in WT animals. Additional morphometric analyses performed using micro-CT approach revealed decreased retina volume and shortened ON in mutant embryos (Fig 1B and C).

### *Tmem107* is highly expressed in the retina during eye development

To investigate the role of *Tmem107* in the development of specific eye structures, we analyzed in situ expression of *Tmem107* using the RNAscope approach during critical stages of eye development (E10–E15). At early stages of optic vesicle outgrowth (E10 and E11) (Fig 2A and B), *Tmem107* expression was detected in the bilayered optic cup, where the inner layer represents the presumptive neural retina (NR) and the outer layer will give rise to the retinal pigment epithelium (RPE). Although *Tmem107* expression was observed

throughout all these structures, the signal was particularly enriched in the presumptive NR (Fig 2A''''–F''''), whereas the lens placode and RPE layer exhibited lower expression (Fig 2A'–F' and A''–F''). The patterns of *Tmem107* expression remained similar during later stages (E12, E13), with the signal located in the anterior and posterior lens epithelium (Fig 2C and D) and in the newly formed cornea (Fig 2C' and D'). At E13, the formation of a ganglion cell layer (GCL) is associated with lower expression of *Tmem107* compared with the rest of NR (Fig 2D''). Later, at E14 and E15, the neuroblast cell layer (NCL), which contains neuronal progenitors, expresses high levels of *Tmem107* (Fig 2E'''' and F''''), whereas differentiated neurons of GCL exhibit low expression of this gene. Interestingly, *Tmem107* mRNA expression in the ciliary marginal zone (CMZ) was very low compared with the rest of the NR (Figs 2F'' and S2A–D). Thus, our findings indicate that *Tmem107* is strongly expressed during early stages of eye development (E10–E15), particularly in the NR.

### Key factors in eye development are altered in $Tmem107^{-/-}$ animals

To gain more insight into molecular changes caused by *Tmem107* deletion, we analyzed in situ expression of key proteins involved in eye patterning. Because the phenotypic analysis revealed distinct anomalies affecting certain eye areas including retina, lens, and optic stalk, we further focused on the evaluation of the expression patterns of proteins that are critical for morphogenesis of the retina (PAX6, SOX2), the optic stalk (PAX2), and the lens (SOX1). PAX6 and SOX2 are transcription factors associated with anophthalmia and microphthalmia in humans (Matsushima et al, 2011). Furthermore, SOX1 is a key regulator expressed in the developing lens (Nishiguchi et al, 1998), and PAX2 has previously been linked to the development of optic stalk and closure of optic fissure (Bosze et al, 2021). Because it is well established that all of these transcription factors are expressed at the early stages of eye development, we performed the analyses at E10.5 and E11.5 (Fig 3). We observed altered expression of PAX6 (Fig 3A–A'' compared with Fig 3B–B''), PAX2 (Fig 3C–C'' compared with Fig 3D–D'') and SOX1 (Fig 3E–E'' compared with Fig 3F–F'') in $Tmem107^{-/-}$ embryos already at E10.5 with more striking differences found at E11.5 (Fig 3G–L). We analyzed the expression of PAX6, PAX2, and SOX1 in microphthalmia mutants because structures expressing these proteins are missing in animals with anophthalmia, which we also see in our mutants. Interestingly, SOX2 expression in microphthalmic $Tmem107^{-/-}$ mutants at E10.5 was still maintained in NR (Fig S3C–C'' compared with Fig S3A–A'') and profound differences were found later at E11.5 (Fig S3F–F''' compared with Fig S3D–D''). In anophthalmic $Tmem107^{-/-}$ mutants, the expression of SOX2 was reduced in the optic stalk, whereas in the optic cup, it was completely lost at E10.5 and E11.5 (Fig S3B'' and E'').

Moreover, the reduction of SOX1 expression was found in the lens of *microphthalmia* mutants, whereas PAX2 and PAX6 were down-regulated in the distal part of NR (Fig 3). In summary, our data indicate that TMEM107 is important for early eye patterning in mice and it may establish the proper expression of pivotal players during crucial stages of the optic cup and stalk morphogenesis.

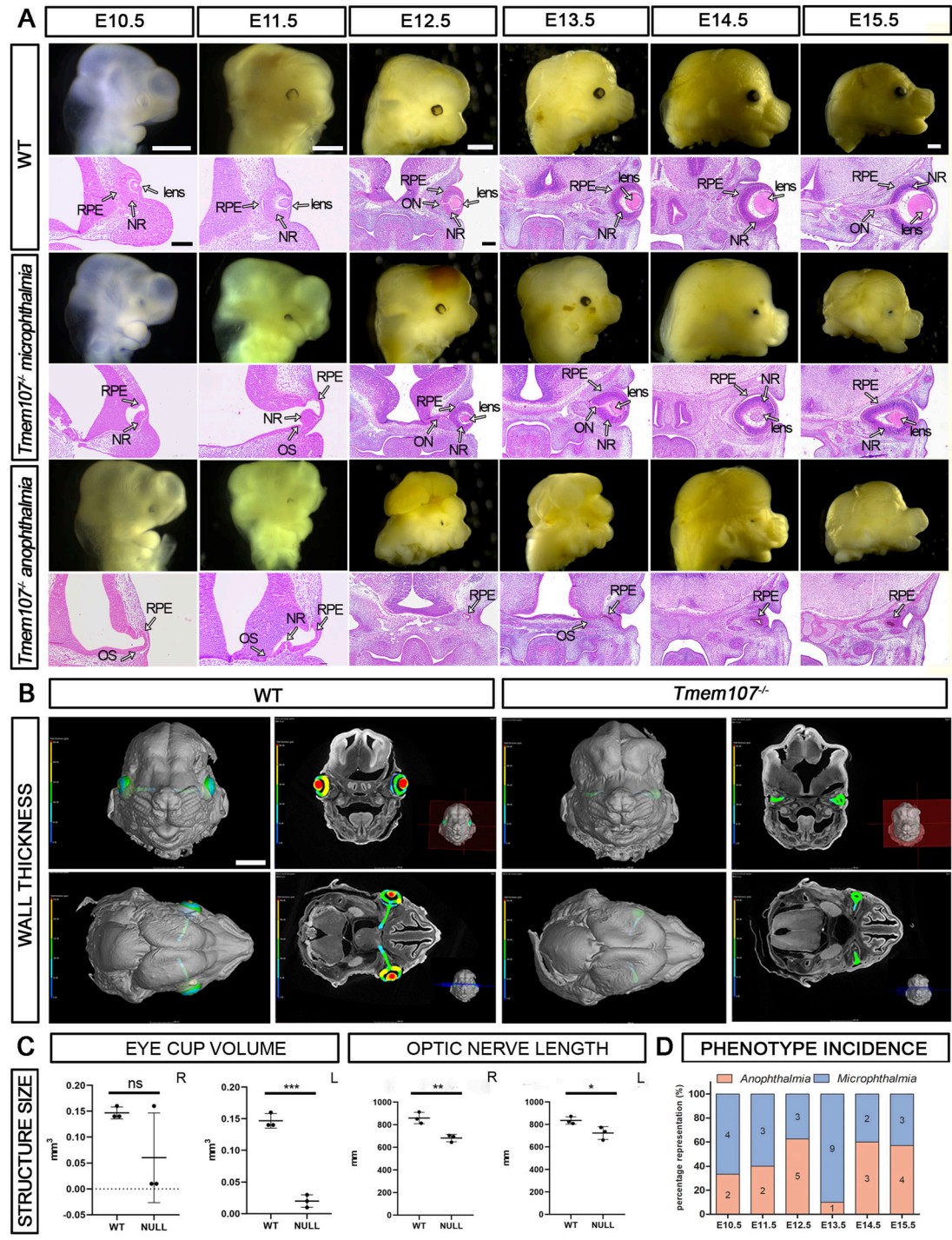

**Figure 1. *Tmem107⁻ᐟ⁻* mutants display severe morphological defects in eye regions.**
**(A)** Macroscopic pictures and HE-stained sections illustrating *microphthalmia* and *anophthalmia* in E10.5–E15.5 mutant embryos compared with WT. **(B)** Micro-CT reconstruction of stage E15.5 WT and *Tmem107⁻ᐟ⁻* eyes. Wall thickness is displayed as a color gradient from blue (the thinnest, 0 μm) to red (the thickest, up to 200 μm) demonstrated in the color legend. **(C)** Graphical representation of changes in eye structure size at stage E15.5 measured from micro-CT scans (left—volume of eyes in mm³; right—optic nerve length in mm) in *Tmem107⁻ᐟ⁻* embryos. Paired, nonparametric, two-tailed $t$ test; ns, nonsignificant; *$P < 0.05$; **$P < 0,01$; ***$P < 0,001$; n = 3. **(D)** Variability of an eye phenotype present in different stages of examined mutant specimens. RPE, retinal pigment epithelium; NR, neural retina; OS, optic stalk; ON, optic nerve. Scale bars: macroscopic pictures = 700 μm; hematoxylin–eosin (HE)-stained sections = 200 μm; micro-CT = 800 μm.

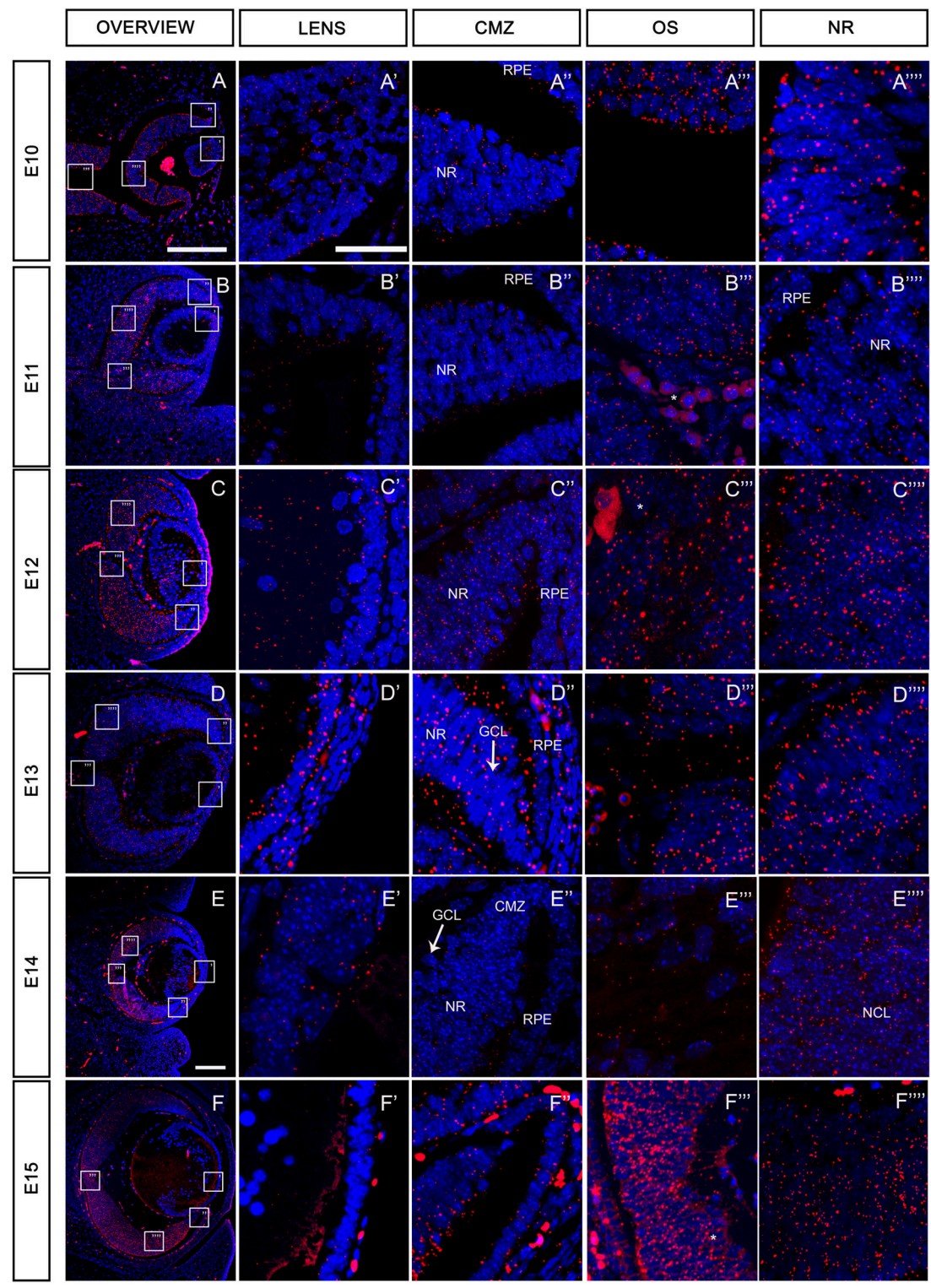

**Figure 2. *Tmem107* expression during eye development.**
**(A, B, C, D, E, F)** Representation of RNAScope signal for *Tmem107* (shown in red) at low magnification in the whole eye at stages from E10 to E15. **(A', A'', A''', A'''', B', B'', B''',**
**B'''', C', C'', C''', C'''', D', D'', D''', D'''', E', E'', E''', E'''', F', F'', F''', F'''')** Details of *Tmem107* expression in lens, (A'', B'', C'', D'', E'', F'') in ciliary marginal zone region, (A''', B''', C''', D''',
E''', F''') in optic stalk/optic nerve region (OS), (A'''', B'''', C'''', D'''', E'''', F'''') and in the neural retina region. Nuclei are counterstained with DAPI (blue). NR, neural retina; RPE,
retinal pigment epithelium; GCL ganglion cell layer; NCL neuroblast cell layer; CMZ, ciliary marginal zone; * autofluorescent blood cells. Scale bar: lower power = 150 *μm*;
higher power = 15 *μm*.

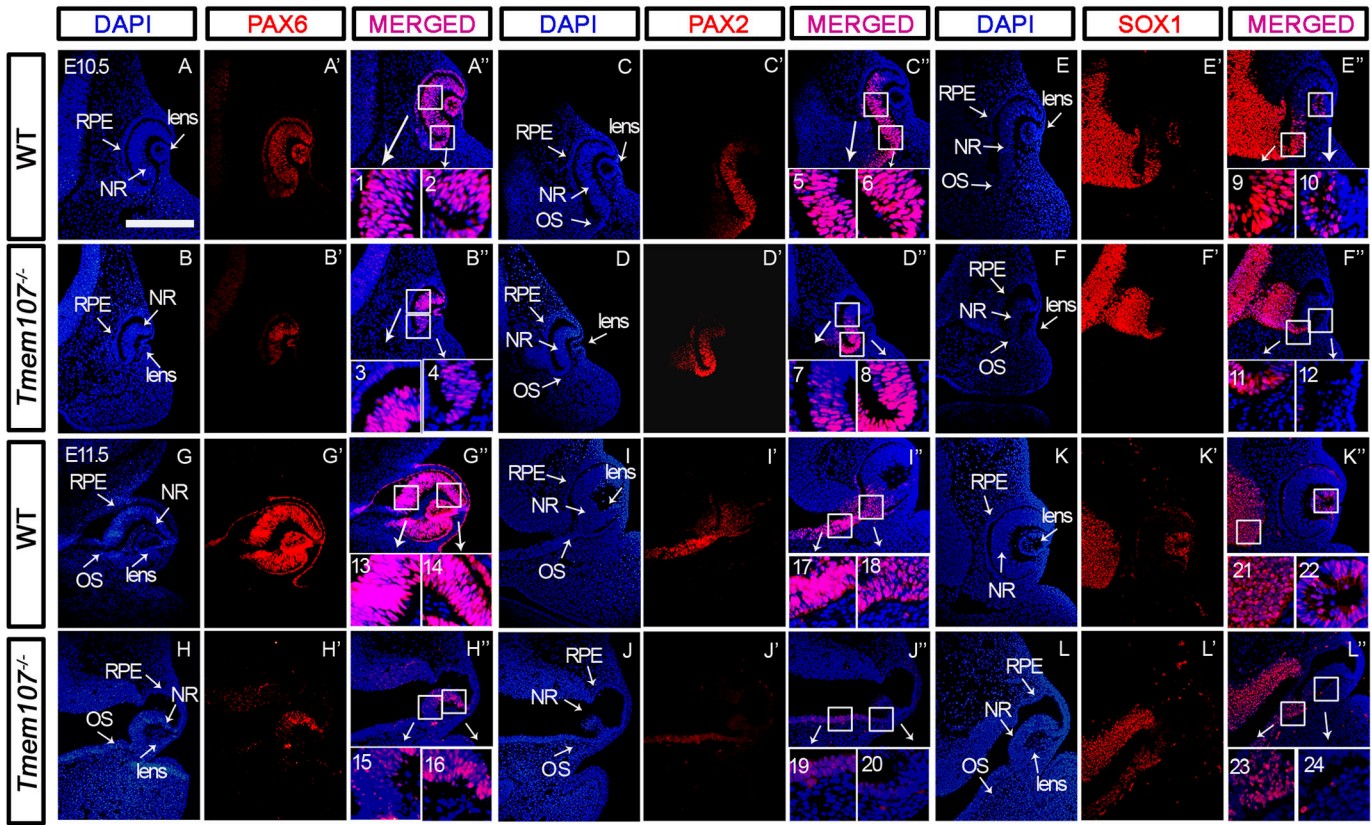

**Figure 3. Key players of eye development are altered in *Tmem107*$^{-/-}$ animals at E10.5–E11.5.**
**(A, A'', A'', B, B', B'', C, C', C'', D, D', D'', E, E', E'', F, F', F'', G, G', G'', H, H', H'', I, I', I'', J, J', J'', K, K', K'', L, L', L'')** Immunohistochemical detection of PAX6 (A, A', A'', B, B', B'', G, G', G'', H, H', H''), PAX2 (C, C', C'', D, D', D'', I, I', I'', J, J', J''), and SOX1 (E, E', E'', F, F', F'', K, K', K'', L, L', L'') proteins in individual eye structures in E10.5 and E11.5 embryos. **(A, A'', A'', B, B', B'', C, C', C'', D, D', D'', E, E', E'', F, F', F'', G, G', G'', H, H', H'', I, I', I'', J, J', J'', K, K', K'', L, L', L'')** PAX6 (red) expression in the neural retina, lens, pigment epithelium, and optic stalk in WT (A, A', A'', G, G', G'') in comparison with *Tmem107*$^{-/-}$ animals (B, B', B'', H, H', H''). PAX2 (red) expression in optic stalk and distal part of neural retina in WT (C, C', C'', I, I', I'') in comparison with *Tmem107*$^{-/-}$ animals (D, D', D'', J, J', J''). SOX1 (red) expression in lens vesicle of WT (E, E', E'') was higher in contrast with *Tmem107*$^{-/-}$ (F, F', F'') similar as in E11.5 WT (K, K', K'') in comparison with *Tmem107*$^{-/-}$ mutants (L, L', L''). Nuclei are counterstained with DAPI. NR, neural retina; RPE, retinal pigment epithelium; OS, optic stalk. Scale bar = 200 $\mu$m.

### TMEM107 deficiency leads to the failure of neural retina formation in human retinal organoids

Given the striking retinal phenotype observed in *Tmem107*$^{-/-}$ animals, we aimed to further test if the human *TMEM107* gene is essential for the development of the retina. We used retinal organoids differentiated from hESCs as a model to closely investigate the roles for *TMEM107* in human retinal development. Moreover, this approach allowed us to evaluate the direct role of *TMEM107* in retinal differentiation without the effects of surrounding or closely associated eye structures and tissues including the brain or surface ectoderm.

We generated *TMEM107*$^{-/-}$ hESCs using CRISPR/Cas9 approach (Figs S4 and S5) and differentiated them into retinal organoids using an already published protocol (Kuwahara et al, 2015; Peskova et al, 2020; Celiker et al, 2023). Retinal organoids were analyzed at day 30 (D30, early stage) and day 150 (D150, late stage) of the differentiation process. Early differentiation steps during the retinal organoid formation include the generation of NR epithelium

containing progenitors that give rise to photoreceptors (*RAX+*), ganglion cells (*VSX2+*), and other cell types of the human retina, whereas the late stage is characterized by photoreceptor maturation (*CRX+*, *RHODOPSIN+*) and appearance of inner and outer photoreceptor segments (Burmeister et al, 1996; Furukawa et al, 1997b; Irie et al, 2015).

At D30 of the differentiation process, WT organoids contained NR epithelium, whereas *TMEM107*$^{-/-}$ organoids lacked NR epithelium and contained cystic structures (Figs 4A and D and S4A–L). RT–qPCR analysis confirmed that *TMEM107*$^{-/-}$ organoids failed to generate NR, as demonstrated by significant down-regulation of genes that are typically expressed in the developing NR structure including *RAX*, *PAX6*, *SOX2*, and *VSX2* (Fig 4B, top row) (Furukawa et al, 1997a; Kozmik, 2008; Matsushima et al, 2011; Burmeister et al, 1996).

Interestingly, scanning electron microscopy (SEM) analysis revealed that *TMEM107*$^{-/-}$ organoids at D30 lack primary cilia on their surface (Fig 4C). In addition, we confirmed the absence of primary cilia using immunofluorescence staining of primary cilia

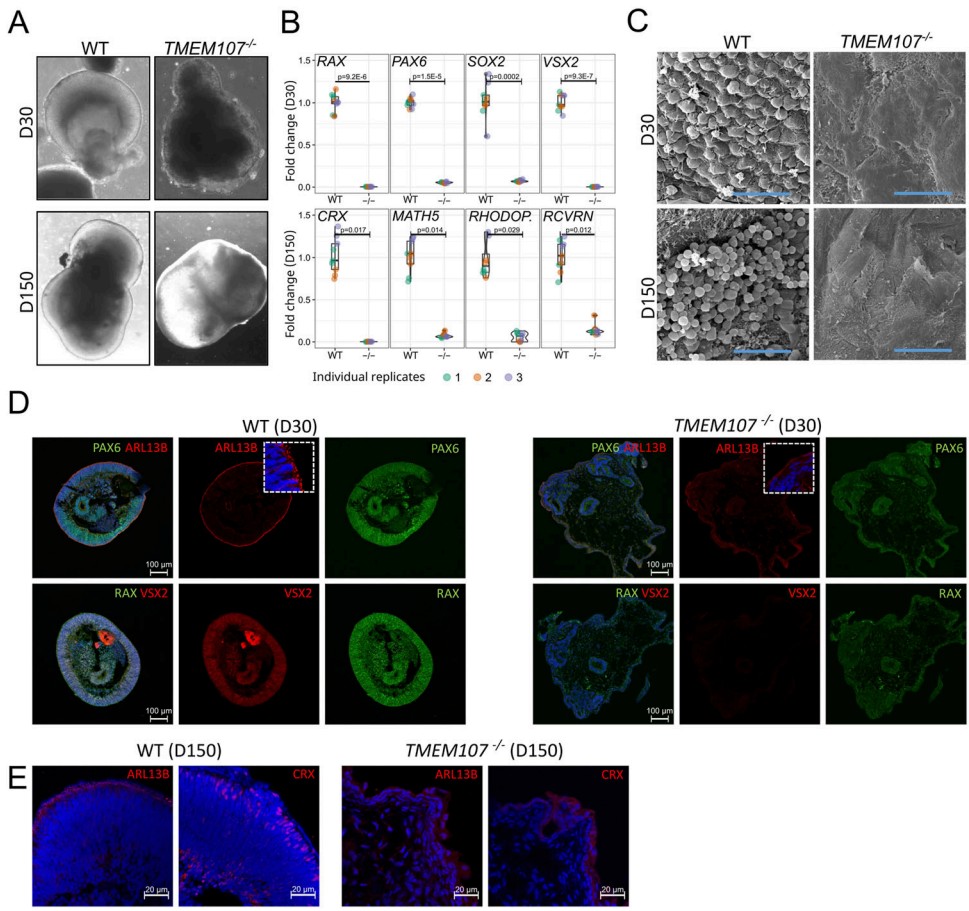

**Figure 4. *TMEM107* is essential for human retinal development.**
**(A)** Morphology of retinal organoids at D30 and D150, as demonstrated using brightfield microscopy. **(B)** Expression of retinal genes at early stage (D30—upper row) and late stage (D150—bottom row) of the differentiation process in WT and *TMEM107*$^{-/-}$ retinal organoids, as demonstrated using RT–qPCR; parametric paired, two-tailed *t* test; n = 3. **(C)** Microphotograph of retinal organoid surface (D30—upper row, D150—bottom row), as demonstrated using SEM. Scale bars represent 10 μm. **(D)** Expression of PAX6 (green), ARL13B (red), RAX (green), and VSX2 (red) in WT and *TMEM107*$^{-/-}$ retinal organoids at D30, as demonstrated using immunofluorescence staining. Nuclei are counterstained with DAPI. **(E)** Expression of ARL13B and CRX in WT and *TMEM107*$^{-/-}$ retinal organoids, as demonstrated using immunofluorescence staining. Nuclei are counterstained with DAPI.

marker ARL13B (Fig 4D). To reveal whether the absence of *TMEM107* leads to the impaired maturation of the retinal organoids and failure to generate photoreceptors and other retinal cell types, we cultured retinal organoids until D150. *TMEM107*$^{-/-}$ retinal organoids at the late stage completely failed to generate NR demonstrated by: (I) altered organoid morphology (Fig 4A), (II) lack of photoreceptor outer segments demonstrated by SEM and immunofluorescence staining for ARL13B (Fig 4C and E), (III) significant down-regulation of photoreceptor markers (*CRX*, *RHODOPSIN*, *RCVRN*) and retinal ganglion cell marker (*MATH5*), as demonstrated by RT–qPCR analysis (Fig 4B, bottom row). Interestingly, Hematoxylin/Eosin staining of retinal organoid cross sections revealed the presence of cysts and Oil Red O lipid staining identified increased the presence of lipids in *TMEM107*-deficient organoids (Fig S4).

To corroborate the phenotype of *TMEM107*$^{-/-}$ organoids, we used a different loss-of-function approach—the shRNA-mediated knockdown of *TMEM107* in human induced pluripotent stem cells (hiPSCs). We generated lentiviral particles containing mCherry reporter and doxycycline (DOX)-inducible expression of shRNA for *TMEM107* down-regulation. Upon transduction, puromycin selection, and FACS sorting, hiPSCs and the generated retinal organoids expressed mCherry reporter (Fig S6A and B). DOX was applied from D2 of the differentiation process, and the retinal organoids were

harvested and analyzed at D25. We found ~50% down-regulation of *TMEM107* gene expression as determined by RT–qPCR in the presence of DOX (Fig S6C) that led to alterations in primary cilia formation including extremely elongated or very short cilia with expanded bulges in their tip (Fig S6D), cyst formation inside of the organoids (Fig S6B), and failure to form NR structures in retinal organoids (Fig S6E), thus corroborating the results generated using the knock-out approach.

Taken together, our results indicate that the absence of *TMEM107* leads to the following: (I) absence of primary cilia on early-stage organoids and outer segments on late stage retinal organoids, (II) the down-regulation of retina-specific genes, (III) the failure to generate the NR structures and cell types in the human retinal organoid model, (IV) the generation of organoid with cysts containing lipids.

### *Tmem107*$^{-/-}$ animals have primary cilia defects in pigment epithelium and neural retina

Primary cilia defect in different organs in *Tmem107*$^{-/-}$ embryos have been previously reported (Cela et al, 2018; Shylo et al, 2020). However, the potential ciliary anomalies in the retina of these animals remain elusive. To test whether *Tmem107*$^{-/-}$ eye phenotype in the mouse model is associated with primary cilia

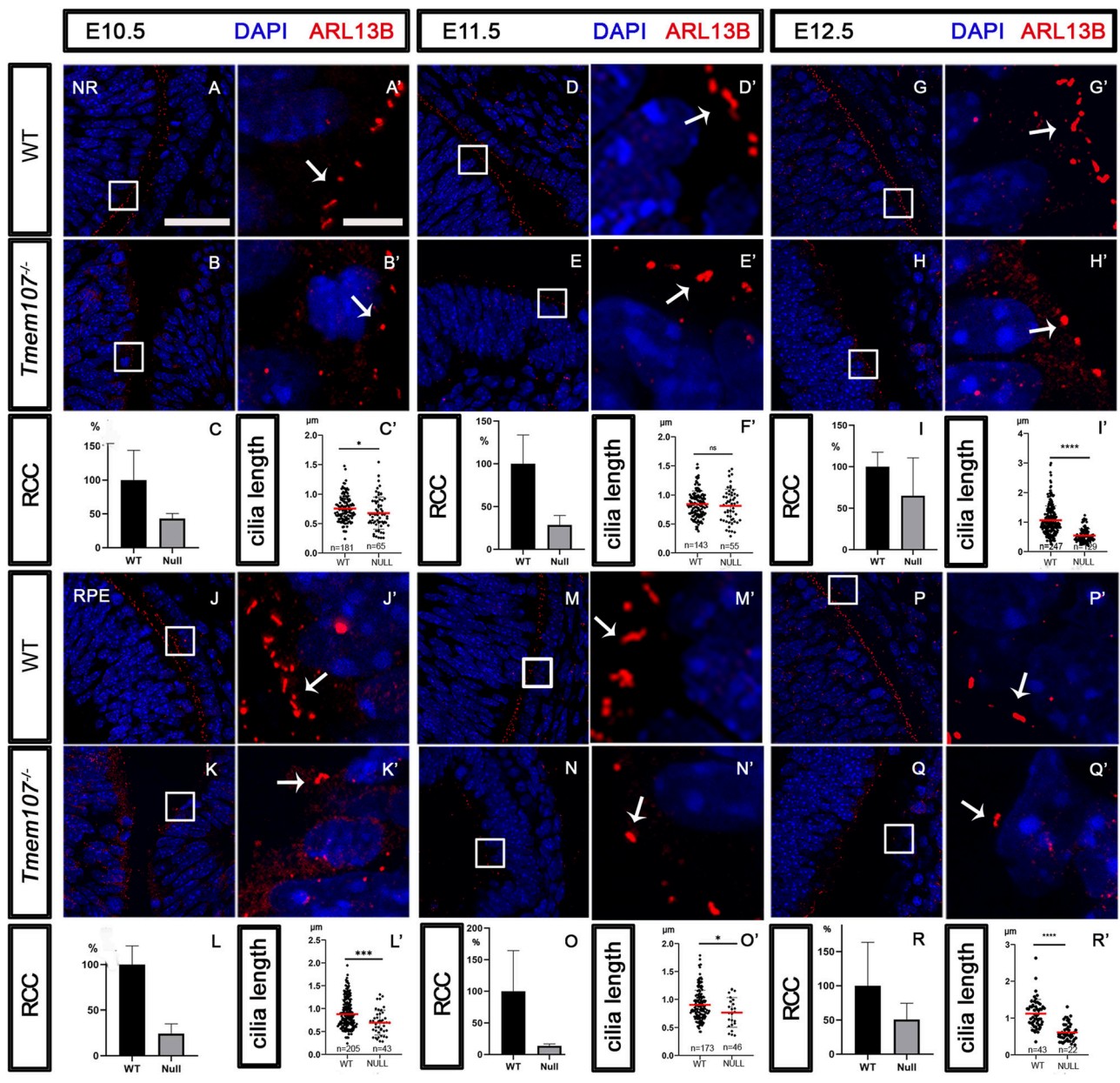

**Figure 5. Primary cilia in neural retina (NR) and retinal pigment epithelium (RPE) Labeling of primary cilia using ARL13B (red) ciliary protein.**
**(A, A´, B, B', D, D´, E, E', G, G´, H, H')** In comparison with WT animals (A, A´, D, D´, G, G´), reduced number and length of primary cilia in NR of E10.5 (B, B'), E11.5 (E, E'), and E12.5 (H, H') $Tmem107^{-/-}$ compared with WT embryos. **(C, C', F, F', I, I')** Graphs representing difference in cilia number in WT (RCC, ratio of ciliated cells) and $Tmem107^{-/-}$ NR (C, F, I) and difference in cilia length ($\mu m$) in WT and $Tmem107^{-/-}$ NR (C', F', I'). **(J, J', K, K', M, M', N, N', P, P', Q, Q')** Comparison of primary cilia in RPE area between WT (J, J´; M, M´; P, P´) and $Tmem107^{-/-}$ display reduced number and length of primary cilia in RPE of E10.5 (K, K'), E11.5 (N, N'), and E12.5 (Q, Q'). **(L, L', O, O', R, R')** Graphs representing difference in cilia number in WT and $Tmem107^{-/-}$ RPE (L, O, R) and difference in cilia length in WT and $Tmem107^{-/-}$ RPE (L', O', R'). Nuclei are counterstained with DAPI. Scale bars: lower magnification = 35 $\mu m$; higher magnification = 5 $\mu m$. n = number of measured cilia; nonparametric unpaired two-tailed $t$ test; ns, nonsignificant; *$P < 0.05$; **$P < 0.01$; ****$P < 0.0001$.

defects, we labeled cilia using anti-ARL13B antibody, counted the number of ciliated cells, and measured the length of primary cilia. We analyzed primary cilia in distinct eye structures with a special focus on RPE and NR at developmental stages E10.5, E11.5 or E12.5. Analysis of the NR region revealed a reduced number of primary cilia (RCC–ratio of ciliated cells) at stages E10.5 (Fig 5A–C), E11.5 (Fig 5D–F), and E12.5 (Fig 5G–I) associated with

significantly reduced cilia length at E10.5 (Fig 5C') and E12.5 (Fig 5I'). In addition, the length of cilia was also reduced at E11.5, but not with the statistical significance (Fig 5F'). However, more profound changes were observed in the RPE region, where we found a decreased length and cilia number in all analyzed samples of different developmental stages: E10.5 (Fig 5J–L'), E11.5 (Fig 5M–O'), and E12.5 (Fig 5P–R'). These data suggest that

TMEM107 plays a role in cilia biogenesis in vivo and regulation of ciliary length in the developing eye.

### Loss of *TMEM107* leads to aberrant SHH signaling in retinal cells

The evident association of primary cilia disruption with eye phenotypes in all our $TMEM107^{-/-}$ models prompted us to proceed with the analysis of the effects of TMEM107 loss at the molecular level. Because the function of the Shh pathway fully relies on the formation of primary cilium (Huangfu & Anderson, 2005), we aimed to closely investigate the effects of TMEM107 loss on this signaling pathway. Retinal organoids do not represent a suitable model to specifically address affected molecular pathways in individual cells, because of the heterogeneity of different cell types they contain, therefore it is challenging to study signaling pathways using an organoid model. We, therefore, used the ARPE-19 cell line, derived from retinal pigmented epithelium, to address the effects of TMEM107 loss on the Shh pathway by generating ARPE-19 $TMEM107^{-/-}$ and $TMEM107^{+/-}$ cell lines, using CRISPR/Cas9 technology.

First, we assessed the expression of the crucial components of the Shh pathway *GLI1* and *PTCH1* using RT–qPCR. Whereas there was no significant difference in *PTCH1* expression between WT and mutant cells, we detected ~15-fold up-regulation of *GLI1* expression in $TMEM107^{-/-}$ cells (Fig 6A), indicating an aberrant activation of the Shh pathway in the absence of *TMEM107*. To confirm the activation of the Shh pathway and to reveal whether ARPE-19 cells possess the functional Shh pathway, we treated the cells using Shh activator smoothened agonist (SAG). Upon SAG treatment, WT and $TMEM107^{+/-}$ cells up-regulated the expression of *GLI1* and *PTCH1* by ~twofold, but $TMEM107^{-/-}$ cells failed to up-regulate *GLI1* and *PTCH1* (Fig 6B and C). Therefore, high *GLI1* expression and no response to SAG treatment of $TMEM107^{-/-}$ cells indicate that the Shh pathway is aberrantly activated in $TMEM107^{-/-}$ cells and that TMEM107 is critical for Shh signaling.

The aberrant Shh signaling in $TMEM107^{-/-}$ cells could be explained by altered primary cilia formation that we, indeed, also observed in $TMEM107^{-/-}$ retinal organoids. To test this, we stained ARPE-19 cells for the ciliary marker ARL13B. WT and $TMEM107^{+/-}$ formed primary cilia, whereas $TMEM107^{-/-}$ completely failed to form these structures (Fig 6D and E). To test SHH activation, we used the SAG treatment approach as described above, and assessed in situ localization of other critical Shh players SMO and GLI2. Immunofluorescence staining revealed that upon SAG treatment SMO localizes into the primary cilia and GLI2 becomes up-regulated and localized into nuclei of WT and $TMEM107^{+/-}$ cells. $TMEM107^{-/-}$ failed to form primary cilia and GLI2 was up-regulated even in the absence of SAG (Fig 6D and E).

Previous studies showed down-regulated SHH in mice with *Tmem107* mutation (Christopher et al, 2012; Cela et al, 2018; Shylo et al, 2020). However, because of distinct ciliary transmission of SHH in different organs, both up-regulation and down-regulation have been observed in different tissues of the developing embryo (Burnett et al, 2017). In models that lack primary cilia, an up-regulation of SHH has been observed in the embryonic retina (Burnett et al, 2017). To test how the altered morphology of primary cilia affected Shh signaling in the mouse model, we analyzed the expression of SHH protein and *Ptch1* RNA at stage E12.5 (Fig 7). Whereas in WT eyes, SHH localized to discrete regions in the distal part of the eye (Fig 7A'), $Tmem107^{-/-}$ animals exhibited higher expression of this ligand in proximal and distal parts of the NR and in optic stalk (Fig 7B). The differences were most prominent in the lens and the optic stalk regions (Fig 7B' and B"). On the contrary, RNAScope analysis of *Ptch1* revealed no distinct changes in its expression in the eyes of E12.5 mutant animals (Fig 7D–D"), as compared with WT (Fig 7C–C"). These data demonstrate that although the expression of SHH was elevated, decrease in primary cilia keeps the pathway dysfunctional in mice mutants.

Taken together, our results indicate that the absence of TMEM107 leads to the following: (I) failure to form primary cilia in retinal pigmented epithelial cells, (II) aberrant up-regulation of the Shh pathway demonstrated by the up-regulation of *GLI1*, *GLI2*, and GLI3 FL, (III) incapability of $TMEM107^{-/-}$ cells to respond to SAG treatment, because lack of cilia, and (IV) increased level of SHH ligands in vivo.

## Discussion

Ciliopathies are a group of genetic disorders characterized by defects in the structure and function of cilia, which are hair-like organelles present on the surface of many cells. Previous research has shown that ciliopathy proteins are important for a variety of developmental processes, including eye development (Waters & Beales, 2011). The protein TMEM107 has been previously implicated in ciliopathy-associated eye abnormalities (Christopher et al, 2012). However, the specific mechanisms by which TMEM107 functions in eye development and the wide range of ocular abnormalities associated with its deficiency have not been fully elucidated.

Here, we used mouse embryos, retinal organoid, and retinal cell culture models to closely investigate the role of TMEM107 in eye development. We found that (I) TMEM107 is specifically and strongly expressed in NR of the developing eye; (II) loss of TMEM107 leads to distinctive ocular phenotypes including anophthalmia and microphthalmia associated with a truncated ON; (III) the expression of crucial genes in eye development is altered in the absence of TMEM107; (IV) TMEM107 is critical for ciliogenesis and Shh signaling, and its absence leads to the disruption of primary cilia and aberrant Shh signaling; and (V) TMEM107 deficiency is associated with the generation of cysts.

All examined $Tmem107^{-/-}$ mouse mutants manifested eye malformations including anophthalmia and microphthalmia. Interestingly, similar phenotypes have been observed in humans. Patients who appear as homozygotes or compound heterozygotes for *TMEM107* mutant allele been diagnosed with Joubert (JS), Meckel–Gruber (MKS) or orofaciodigital syndrome (OFD) (Iglesias et al, 2014; Shaheen et al, 2015; Lambacher et al, 2016; Shylo et al, 2016; Chinen et al, 2022). All abovementioned syndromes have been recognized as ciliopathies associated with eye defects like *anophthalmia*, *microphthalmia*, retinal defects, coloboma or lid anomalies (Hartill et al, 2017; Hartill et al, 2017). Patients with *TMEM107* pathological

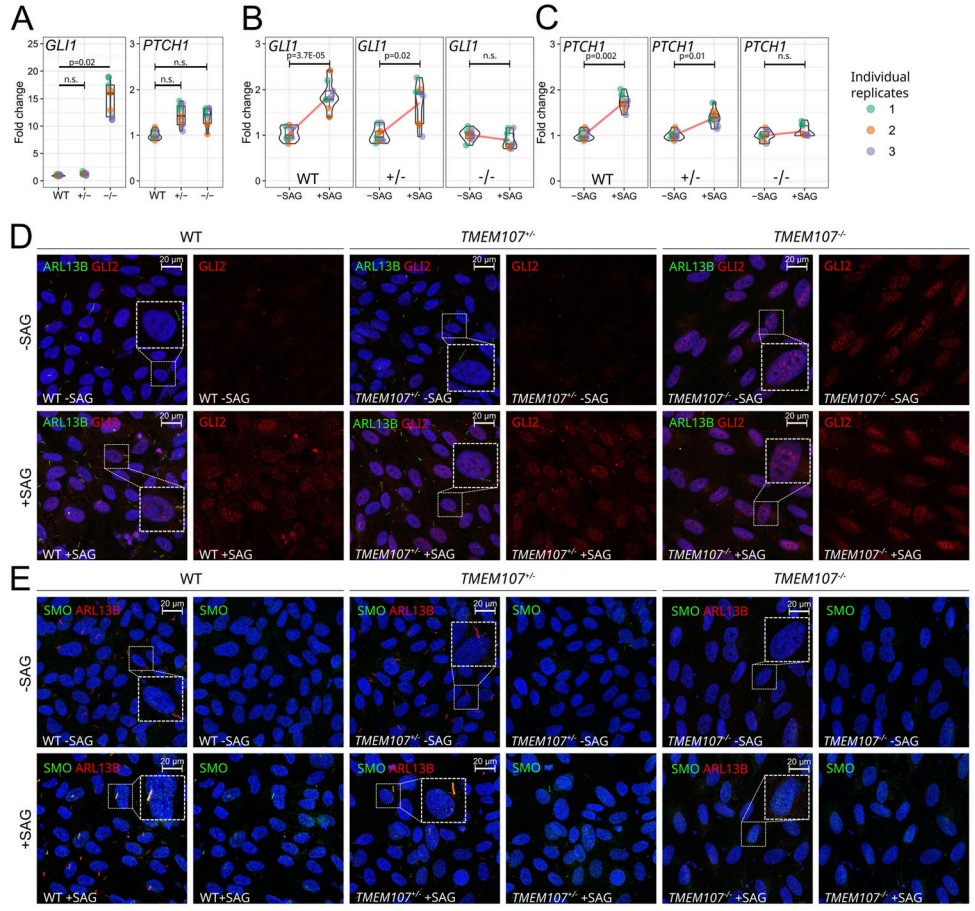

**Figure 6.   Loss of *TMEM107* leads to aberrant Shh signaling in retinal cells.**
**(A)** Expression of *GLI1* and *PTCH1* in WT, *TMEM107*⁺/⁻, and *TMEM107*⁻/⁻ ARPE-19 cells, as demonstrated using RT–qPCR; Paired, two-tailed *t* test; n.s. = non-significant; n = 3. **(B, C)** Expression of *GLI1* and *PTCH1* upon SAG treatment in WT, *TMEM107*⁺/⁻, and *TMEM107*⁻/⁻ ARPE-19 cells, as demonstrated using RT–qPCR; Paired, two-tailed *t* test; n.s., nonsignificant; n = 3. **(D)** Expression of ARL13B (green) and GLI2 (red) upon SAG treatment in WT, *TMEM107*⁻/⁻, and *TMEM107*⁺/⁻ ARPE-19 cells, as demonstrated using immunofluorescence staining. **(E)** Expression of ARL13B (red) and SMO (green) upon SAG treatment in WT, *TMEM107*⁻/⁻, and *TMEM107*⁺/⁻ ARPE-19 cells. Nuclei are counterstained with DAPI (blue).

variants were diagnosed with a stronger phenotype in case of MKS with bilateral *anophthalmia* (Shaheen et al, 2015), and milder phenotypes with OFD and JBT with oculomotor apraxia and retinopathy (Lambacher et al, 2016), and OFD with strabismus (Chinen et al, 2022). Severity of exhibited symptoms seems to be correlated with the type of patients´ mutations. In the case of two OFD patients, the sequencing analysis detected a homozygous missense variant, whereas in the Joubert syndrome patient, a compound heterozygous mutation containing a frameshift deletion and an in-frame deletion was discovered (Lambacher et al, 2016). On the other hand, the patient carrying one intronic base pair insertion causing frameshift and premature protein truncation developed more severe defects (Shaheen et al, 2015). Similarly, in ciliopathic mice, different eye defects have been described (Burnett et al, 2017; Fiore et al, 2020). Homozygotic hypomorphic *Tmem107*^schlei^ mouse embryos develop milder phenotype–*microphthalmia* (Christopher et al, 2012), whereas mutant mice used in this study exhibit more severe eye defects when compared with those containing hypomorphic alleles. We have shown that all of the examined mutants manifest eye malformations with the most severe form—*anophthalmia* (33% mutants). One of the reasons why all of the examined *Tmem107*-deficient embryos develop severe phenotypes whereas milder defects are observed in humans could be because of differences between species and the fact that our models exhibit total lack of TMEM107

protein. Although mutations occurring in humans could lead to expression of a truncated version of the protein, which still partially retains the function, the complete loss of the protein may lead to more severe consequences. Another fact to be considered is that human embryos carrying *TMEM107* mutations die at early pregnancy and are not being diagnosed. Interestingly, *Tmem107*⁻/⁻ mice display ON hypoplasia, which can also occur in patients with Meckel–Gruber syndrome (MacRae et al, 1972). Furthermore, these mutants were shown to have other defects such as *exencephaly*, polydactyly, and cleft palate, which are observed in human *TMEM107* homozygotic patients and are also often present in other ciliopathic cases (Waters & Beales, 2011). Animals carrying only one mutated *Tmem107* allele do not exhibit any obvious phenotype which corresponds to findings on human patients' parents who do not develop any of the symptoms present in children (Cela et al, 2018).

Given the striking eye phenotype in *Tmem107*⁻/⁻ mouse embryos and the lack of current knowledge about in situ *Tmem107* expression in developing eye structures, we aimed to assess *Tmem107* expression during eye development. We found an elevated *Tmem107* expression in the presumptive NR. High *Tmem107* expression in NR and *micro-/anophthalmia* phenotypes in the absence of this gene indicates a critical role for *Tmem107* in the development of the NR. Because eyes develop as brain evaginations, one could note that the observed eye phenotypes could be a consequence of not properly

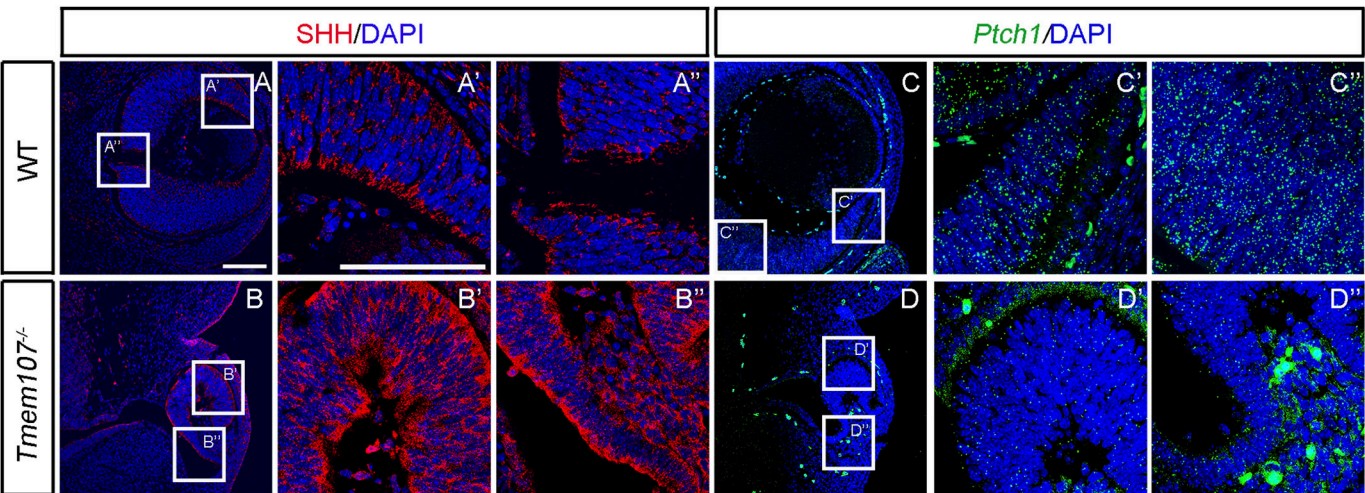

**Figure 7. Expression of HH pathway members in E12.5 *Tmem107*⁻/⁻ animals.**
**(A, A', A'', B, B', B'')** Immunostaining of WT embryos (A, A', A'') and *Tmem107*⁻/⁻ (B, B', B'') revealed higher expression of SHH (red) protein in neural retina of *Tmem107*-deficient animals. **(C, C', C'', D, D', D'')** WT embryos (C, C', C'') exhibits revealed higher expression of *Ptch1* (green) mRNA in lens and neural retina in *Tmem107*⁻/⁻ (D, D', D'') as analyzed by RNAScope. Nuclei are counterstained with DAPI. Scale bar = 100 μm.

developed brain rather than "stand alone" phenomena. To address this question, we performed 3D in vitro experiments using a human retinal organoid model. The retinal organoid system provided us not only the opportunity to examine the effects of *TMEM107* loss on retinal development without any possible influence of other tissues and structures, but also to study the role of this gene in humans. Similarly, as in the mouse model, retinal organoids generated from *TMEM107* knock-out hESCs were unable to form the NR.

Early morphogenesis of the eye is characterized by distinctive spatial and structure specific expression of different transcription factors. SOX2 expression is detected in presumptive eyes from the early developmental stages (Kamachi et al, 1998) and has multiple roles during eye maturation. Importantly, it has been demonstrated that retinal progenitor cells with conditionally ablated SOX2 lose competence to both proliferate and terminally differentiate (Taranova et al, 2006). Interestingly, we found profound differences of SOX2 expression between anophthalmic and microphthalmic mutants at E11.5 (Fig S3). This could be explained by the different molecular mechanisms driving the expression of SOX2 at different stages of eye development and as a consequence of losing SOX2 expression in the optical cup leading to inability of retinal cells to proliferate and differentiate in the absence of TMEM107 protein. SOX1 is a transcription factor that plays an important role in eye development. During early stages, SOX1 is expressed in the optic stalk, presumptive retina, and lens, whereas at later stages, its expression is restricted to lenses where it plays a critical role in γ-*crystallin* production (Nishiguchi et al, 1998). In *Tmem107*⁻/⁻ mutants, we observed a loss of SOX1 in the lens area, whereas presumptive retina and optic stalk remain positive at E10.5 (Fig 3). TMEM107-deficient human retinal organoids largely support this scenario and do not express all these transcription factors or in different extents (Fig S6).

Previous analysis of *Tmem107*⁻/⁻ animals revealed reduced ciliogenesis in mesenchymal cells associated with altered morphology of primary cilia (Cela et al, 2018; Shylo et al, 2020). These findings corroborated the results previously reported in *Tmem107*^schlei mice

and in fibroblasts derived from human patients (Christopher et al, 2012; Shaheen et al, 2015; Shylo et al, 2016). Similar to already published studies, our data revealed profoundly reduced ciliogenesis in NR and RPE cells in mice mutants. However, the length of primary cilia was significantly reduced in all eye regions examined, which is in agreement with our observations in the nodal cilia of Tmem107 null animals (Shylo et al, 2020), however in contrast with the previous findings showing elongation of some primary cilia in fibroblasts of both *Tmem107*^Null and *Tmem107*^schlei mutants (Christopher et al, 2012; Cela et al, 2018) and on fibroblasts derived from patients (Shaheen et al, 2015; Shylo et al, 2016). The distinct ciliary phenotype in different tissues indicates tissue-specific requirements of TMEM107 for ciliogenesis. This might be even more specific for the eye area with the elevated TMEM107 expression.

Importantly, *TMEM107*⁻/⁻ organoids form cysts containing lipids and accumulated liquid inside. Cysts are typically formed in numerous organs of ciliopathic patients, including polycystic kidneys (Wilson, 2004). Liquid accumulation during the cyst initiation is usually caused by the altered function of gated channels, commonly present in ciliary membrane and at the ciliary base bordering with the TZ, where TMEM107 is localized (Gallagher et al, 2006; Slaats et al, 2015). For example, in the *retinoschisis* organoid model, a protein NPHP5 that typically localizes to the TZ was down-regulated resulting in reduced ciliogenesis and generation of congenital retinal cysts by splitting of inner layers of the peripheral retina (Tantri et al, 2004; Huang et al, 2019). Importantly, the loss of adhesion in the cystic retinal organoid model also represents a common mechanism of renal cyst development (van Adelsberg, 2000; Roitbak et al, 2004; Huang et al, 2019). Interestingly, despite the fact that neural retina is not properly developed in *Tmem107*-deficient animals and retinal organoids, small pigment residues are still formed in both models. This is fully in line with the current understanding of eye development, as RPE differentiation is regulated by Wnt signaling that does not fully rely on a functional

primary cilium (Eiraku et al, 2011), although some crosstalk between Wnt and SHH pathways has been documented (Wiegering et al, 2019).

SHH pathway is known to regulate ocular development in humans and mice (Cavodeassi et al, 2019). Anophthalmic and microphthalmic eye phenotypes and other defects in $Tmem107^{-/-}$ mice suggest an altered activation of the SHH pathway. In fact, we found elevated expression of SHH ligand despite a reduced number of primary cilia in Tmem107-deficient animals. This implies the possibility of ciliary involvement in the negative regulation of Shh gene expression or altered functionality of the remaining primary cilia. Indeed, the role of SHH in the regulation of dorso-ventral patterning of an optic cup has been also described (Zhao et al, 2010). SHH induces Pax2 expression, PAX2 then binds the α-enhancer to antagonize Pax6 expression in the ventral optic vesicle (Chiang et al, 1996; Heavner & Pevny, 2012). Here, we have shown down-regulated PAX6 in proximal optic cup regions at early developmental stages (E10.5 and E11.5), which is probably a consequence of elevated SHH levels in the absence of primary cilia. Moreover, it points to the fact that microphthalmia in $Tmem107^{-/-}$ animals is likely caused by down-regulated Pax6, whereas anophthalmia is rather caused by altered expression of Sox2. Nevertheless, our analyses of differentiation stages revealed a strong down-regulation of Pax6 in the retina, which might be the consequence of partially restored function of remaining cilia in the retina or as a result of interrupted communication between developing tissues in the eye.

Recent studies have determined the association between primary cilia and early eye development. Mouse models lacking crucial ciliary proteins have aberrant or no primary cilia, exhibit patterning defects, and abnormal eye morphology induced by ectopic activation of SHH signaling (Burnett et al, 2017). Furthermore, loss of ciliary gene Arl13b in mice produced a distinct eye phenotype characterized by an inverted optic cup (Fiore et al, 2020). These eye phenotypes were associated with aberrant SHH signaling and were rescued by Gli2 deletion (Burnett et al, 2017; Fiore et al, 2020). We observed aberrantly activated Shh pathway with up-regulated SHH in mice and up-regulated GLI1/2 in human ARPE-19 cells. In addition, because of the aberrantly activated Shh pathway, $TMEM107^{-/-}$ human cells are not able to further up-regulate Shh downstream molecules. These data suggest a mechanism for Shh signaling in eye development, whereby eye development is probably driven by the repressive Gli transcription factor signaling, rather than their transcription-activating roles.

Thus far, all described stem cells were found to have primary cilia (Yanardag & Pugacheva, 2021). The primary cilium displays a critical role in regulating stemness and primary cilia-dependent signaling is required for MSCs' proliferation and pluripotency (Yanardag & Pugacheva, 2021). Previous research has shown correlation between high doses of Sox2 and repression of proliferative genes in slow-cycling stem cells (Hagey & Muhr, 2014) and the expression of stem cell marker Sox2 is down-regulated after the deciliation (Ma et al, 2020). In contrast, upon differentiation, reduction of Sox2 levels is needed for cyclin D1-dependent entrance of rapidly dividing states (Hagey & Muhr, 2014). Furthermore, ciliary disassembly is needed to release brakes on G1/S transition for cells to enter into a fast proliferating state (Izawa et al, 2015). Our data reveal overlapping expression patterns of

Tmem107 and Sox2 genes in NR (Figs S2 and S3). Both of them are abundantly present in pluripotent neuroepithelial cells and down-regulated in differentiating GCL cells (Taranova et al, 2006; Zhang & Cui 2014). Furthermore, we have shown positive correlation between Tmem107 absence and SOX2 down-regulation, suggesting that SOX2 role in maintaining pluripotency could be directly regulated by TMEM107. Interestingly, CMZ exhibits no expression of SOX2 together with very low expression of Tmem107 (Fig S2). This area generates new neurons that are incorporated into the retina as it enlarges continually throughout the lifetime of an animal (Harris & Perron, 1998; Reh & Levine, 1998). This peripheral area of the adult eye may also contain retinal stem cell pools (Ahmad et al, 2000; Tropepe et al, 2000; Coles et al, 2006). These findings further confirm a key role of TMEM107 in differentiation of neural retina and also suggest that SOX2 regulation specifically requires TMEM107.

In conclusion, our results enhance the understanding of developmental processes in ciliopathies where ocular phenotypes occur and help us gain deeper knowledge into the mechanisms involved in primary cilia function during morphogenesis of eye structures. Our findings give insight into the relationship between primary cilia-mediated signaling and eye development, revealing reduced ciliogenesis and aberrant Shh signaling as the cause of anophthalmic and microphthalmic phenotypes and could expand diagnostic tools for these diseases in humans by classifying Tmem107 as one of the diagnostic markers. Furthermore, we introduce retinal organoids as a useful tool to study mechanisms of primary cilia disruption and for the testing of potential therapeutic approaches to treat ciliopathy-related diseases.

# Materials and Methods

## Animals

Embryos of mice containing $Tmem107^{null}$ allele were generated as described previously (Tang et al, 2010; Christopher et al, 2012). The details on the generation of the mutant can be found via the MMRRC strain information page (https://www.mmrrc.org/catalog/sds.php?mmrrc_id=32632). Targeted or gene trap mutations were generated in strain 129/SvEvBrd-derived embryonic stem (ES) cells. The chimeric mice are bred to C57BL/6J albino mice to generate F1 heterozygous animals. All of the mouse husbandry was performed in accordance with Yale's Institutional Animal Care and Use Committee guidelines.

## Immunofluorescent staining on paraffin-embedded sections of mice

Embryonic mouse samples were collected (Table S2) and fixed in 4% paraformaldehyde overnight and then processed for paraffin embedding and sectioned in transversal planes. Before immunofluorescent analysis, sections underwent the xylene/ethanol series for deparaffinization and rehydration. For antigen retrieval, 0.1 M sodium-citrate buffer pH6 with 0.04% Tween was used. Retrieval was performed under 98°C for 25 min.

Sections were incubated with primary antibodies (Table S3) for 1 h at room temperature and subsequently with the following secondary antibodies for 1 h at room temperature: anti-mouse Alexa Fluor 488 (1:200, cat. No. A11001), anti-mouse Alexa Fluor 568 (1:200, cat. No. A11004), anti-rabbit Alexa Fluor 488 (1:200, cat. No. A11008), and anti-rabbit Alexa Fluor 594 (1:200, cat. No. A11037; all Thermo Fisher Scientific). Sections were washed in PBS and coverslipped in Fluoroshield with DAPI (cat. No. F6057; Sigma-Aldrich, Merck). The pictures were taken under a fluorescence microscope Leica DM LB2 (Leica) and the Leica SP8 confocal microscope (Leica). Individual images were assembled into Figures in Adobe Photoshop 5.0. Pictures of wild-type and mutant specimens were taken under the same microscope settings.

## Immunofluorescent staining of organoids

For frozen sections staining, organoids were fixed in 4% paraformaldehyde for 30 min at room temperature, washed with PBS, cryopreserved in 30% sucrose (16104; Sigma-Aldrich) in PBS overnight at 4°C, embedded in Tissue-Tek O.C.T. Compound medium (Sakura) and sectioned on a cryostat (CM1850; Leica) with thickness of slides ~10 $\mu$m. For cells grown on coverslips, cells were fixed in 4% paraformaldehyde for 15 min at room temperature. Samples were washed with PBS and blocked in a blocking buffer (PBS with 0.3% Triton X-100 [T8787; Sigma-Aldrich] and 5% normal goat serum [G9023; Sigma-Aldrich]) for 1 h at room temperature in a humidified chamber. The primary antibodies were diluted in antibody diluent (PBS with 0.3% Triton X-100 and 1% bovine serum albumin [A9647; Sigma-Aldrich]) and applied to the sections overnight at 4°C in a humidified chamber. List of primary antibodies is provided in Table S3. The sections were washed with antibody diluent and a secondary antibody (anti-Mouse AlexaFluor 488; Thermo Fisher Scientific, 1:1,000 and anti-rabbit AlexaFluor 594; Thermo Fisher Scientific, 1:1,000) was applied for 1 h at room temperature in a humidified chamber. The nuclei were stained with 1 $\mu$g/ml DAPI in PBS for 4 min at room temperature and the sections were mounted using Fluoromount Aqueous Mounting Medium (F4680; Sigma-Aldrich). Sections were scanned using Zeiss LSM 880 confocal microscope and files were processed using ZEN Blue Edition software (Zeiss).

## In situ gene expression analysis using RNAScope technique

The embryos were fixed in 4% PFA for 24–36 h and proceeded for dehydration in an ethanol series. After embedding in paraffin, 5-$\mu$m transverse sections were obtained. The sections were deparaffinized in xylene and dehydrated in 100% ethanol. To detect mRNA expression of *Tmem107* (805841; ACD Bio) and *Ptch1* (476231; ACD Bio) in mouse embryos, we used the RNAScope R Multiplex Fluorescent v2 Assay kit (Cat. No. 323 110; ACD Bio) for formalin-fixed paraffin-embedded tissues according to the manufacturer's instructions. All reactions were performed at 40°C in the HybEZTM II Oven (ACD Bio). Visualization of hybridized probes was done using the TSA-Plus Cyanine 3 system (NEL744001KT; Perkin-Elmer), according to the manufacturer's protocol. DAPI (cat. n. 323 108; ACD Bio) was used for nuclear staining. Pictures were taken with a

fluorescence microscope Leica DM LB2 (Leica) and the Leica SP8 confocal microscope (Leica).

## CRISPR/Cas9-mediated gene editing

Cells were transfected with a PX458 vector containing gRNA sequences for *TMEM107* knockout. The pSpCas9(BB)-2A-GFP (PX458) was a gift from Feng Zhang (Plasmid #48138; Addgene; http://n2t.net/addgene:48138; RRID:Addgene_48138). gRNA was designed using Benchling [Biology Software] (2021), retrieved from https://benchling.com, to target the *NcoI* restriction site (Fig S7A) in the first exon of *TMEM107* (FWD: CACCGAAGCCCTGAGACCCGGCCCA, REV: aaacTGGGCCGGGTCTCAGGGCTTC). 48 h after transfection, GFP-positive cells were bulk FACS-sorted using BD FACS Aria. Then, the cells were transfected for the second time and individual clones were generated using single-cell FACS sorting. Selected clones were screened by PCR using the following primers flanking the first *TMEM107* exon (FWD: CTAGATTTGTCGGCTTGCGG, REV: GCAGGGTAGGACTAGGACCA). PCR products were digested using the NcoI restriction enzyme (Fig S7B and C). Clones that lacked the *NcoI* restriction site were selected for the final next-generation sequencing screen using the Illumina platform (Malcikova et al, 2015) and compared with the wild-type sequence (Figs S8 and S9).

## shRNA-mediated gene knockdown

DNA sequences for shRNA-mediated *TMEM107* down-regulation were designed using a web tool (Gu et al, 2014). Generated sequences: shRNA-TMEM107-F: CCGGGACATTAAGACTTATATAAtaacctgacccattaTTA-TATAAGTCTTAATGTCTTTTTG, shRNA-TMEM107-R: AATTCAAAAAGA-CATTAAGACTTATATAAtaatgggtcaggttaTTATATAAGTCTTAATGTC were cloned into a lentiviral vector containing DOX-inducible U6 promoter and TetRep-P2A-Puro-P2A-mCherry (Eshtad et al, 2016) (vector kindly provided by Mikael Altun). Lentiviral particles were generated as described previously (Barta et al, 2016; Peskova et al, 2019, 2020) using pMD2.G (#12259; Addgene) and psPAX2 (#12260; Addgene) (gift from Didier Trono, École Polytechnique Fédérale de Lausanne, Lausanne, Switzerland). We generated non-clonal hiPSC lines using fluorescent activated cell sorting of mCherry-positive hiPSCs. After transduction, at least 5,000 mCherry-positive hiPSCs were sorted using BD FACSAria (BD Biosciences). Generated cell lines were then propagated in the presence of puromycin (1 $\mu$g/ml). DOX 0.5 $\mu$g/ml was applied from day (D)2 to D25 of the differentiation of retinal organoids.

## Analysis of primary cilia length

Ciliary axonemes were visualized using ARL13B immunostaining as described above. Imaging was performed with a Leica SP8 laser scanning microscope using the 63× oil objective, and image acquisition was performed using LAS X software (Leica). Image data were acquired as Z-stacks of images and a 0.3-$\mu$m distance separated neighboring Z-sections. Cilia length measurements were done using the Imaris Cell module (Imaris v9.5; Oxford Instruments) using the methodology of fitting an ellipsoid into the surface analysis to obtain ellipsoid axis length. Statistical analyses were performed using a two-tailed *t* test.

## Generation of retinal organoids

Retinal organoids were generated from human embryonic stem cells (hESCs) (H9 cell line, WiCell Bank) or hiPSCs (M8 cell line, [Peskova et al, 2020]), according to the protocol described elsewhere ([Kuwahara et al, 2015]; [Peskova et al, 2020]) with small modifications. Briefly, cells in E8 medium were seeded into a U-shaped, cell-repellent 96-well plate (5,000 cells/well) (Item No: 650970; Cellstar, Greiner) in the presence of 20 $\mu$M ROCK inhibitor (Y-27632; STEMCELL Technologies). After 48 h (Day 0 of the differentiation process) the culture medium was changed to gfCDM ([Kuwahara et al, 2015]) (45% IMDM [Item No: 21980-032; Gibco], 45% Hams F12 [F12, Item No: 21765-029; Gibco], 10% KnockOut Serum Replacement [Item No: 10828-028; Gibco], 1% chemically defined lipid concentrate [Cat. No. 11905-031; Gibco], 1× penicillin–streptomycin solution [Item No: XC-A4122; Biosera], 10 $\mu$M $\beta$-mercaptoethanol [Item No: M3148; Sigma-Aldrich]). On day 6, recombinant human BMP4 (Peprotech) was added to the culture to the final concentration 2.2 nM and then the medium was changed every third day. On day 18 of the differentiation process, gfCDM was changed to a NR medium (DMEM/F12 [Item No: 21331-020; Gibco], 1% N-2 supplement [Item No: 17502001; Gibco], 1% GlutaMAX supplement [Item No: 35050061; Gibco], 10% FBS [Item No: FB-1101; Biosera], 0.5 $\mu$M retinoic acid [Item No: R2625; Sigma-Aldrich], 0.1 mM taurine [Item No: T8691; Sigma-Aldrich], 1× penicillin–streptomycin solution [Item No: XC-A4122; Biosera]).

## Cell cultures and their treatments

ARPE-19 (ATCC - American Type Culture Collection) cells were cultured in media Knockout DMEM (Invitrogen, Life Technologies Ltd.), containing 10% FBS, 1× GlutaMAX (Invitrogen, Life Technologies Ltd.), 1× MEM nonessential amino acid solution (Invitrogen, Life Technologies Ltd.), 1× penicillin/streptomycin (PAA), and 10 $\mu$M $\beta$-mercaptoethanol (Sigma-Aldrich). The cells were incubated at 37°C/5% $CO_2$ and regularly passaged using trypsin (Sigma-Aldrich). For the small-molecule agonist of the Shh-Smo pathway (SAG) treatment, confluent ARPE-19 cells were cultured 24 h in the presence of 1 $\mu$M SAG (Tocris) without FBS and harvested or fixed for downstream processing. H9 hESCs and M8 hiPSCs were cultured in Essential 8 Medium (Invitrogen, Life Technologies Ltd.) on cell culture dishes coated with vitronectin and regularly passaged using 0.5 mM EDTA in PBS.

## Gene expression analyses by RT–qPCR

ARPE-19 cells were washed with PBS and harvested into 1 ml of RNA Blue Reagent (an analog of Trizol) (Top-Bio). For retinal organoid analysis, at least eight organoids were washed with PBS and homogenized using a 1 ml insulin syringe in 1 ml RNA Blue Reagent. RNA was isolated using chloroform extraction, and then reverse-transcribed using a High-Capacity cDNA Reverse Transcription Kit (Applied Biosystems). The RT product was amplified by real-time PCR (Roche LightCycler 480 PCR instrument) using PowerUp SYBR Green Master Mix (Applied Biosystems TM). Primer sequences are

shown in Table S4. All experiments were performed using at least three independent replicates.

## Micro-CT analyses

Before micro-CT scan the samples of E15.5 mouse embryos were dehydrated in the alcohol grades (30%, 50%, 70%, 80%, and 90%). The samples were then stained in the solution of 1% iodine in 90% MeOH to enhance the contrast of soft tissue. For the purpose of motion stabilization during the scan, the embryos were embedded in 1% agarose gel in a 2-ml Eppendorf tube.

The micro-CT scanning was performed using laboratory system GE phoenix v|tome|x L 240 (Waygate Technologies), equipped with a 180 kV/15W maximum power nanofocus X-ray tube and high-contrast flat panel detector dynamic 41|100 (dimension - 4,048 × 4,048 pixels, with pixel size of 100 $\mu$m). The measurements were carried out in an air-conditioned cabinet (21°C). The tomographic reconstruction was performed using the software GE phoenix datos|x 2.0 (Waygate Technologies, Germany). The measurements were acquired with the following settings: acceleration voltage 60 kV, X-ray tube current 200 $\mu$A, exposition time was 900 ms, 2,200 images were taken over the 360° rotation, and voxel resolution of 3.6 $\mu$m.

The manual segmentation of the eyes was carried out using Avizo 9.5 software (Thermo Fisher Scientific). All 3D visualizations, measurements, and specialized analyses were performed in VG Studio MAX 2022 software (Volume Graphics GmbH).

The wall thickness analysis was performed in the VG Studio software. The analysis is based on fitting the spheres in the 3D space into the defined object of the eye. The larger the space in between the walls of the object in 3D can fit a larger sphere. The quantification of this analysis is showing the count of the voxels (voxel = 3D pixel) with the appointed wall thickness.

The optic stalk length was measured with the polyline tool. This tool enables the length measurement of the not-straight objects by placing the individual points defining the line alongside the shape of the measured object. The volumes of the segmented eyes were measured in the VG Studio software.

## SEM

SEM was performed as described previously ([Peskova et al, 2020]). Briefly, the organoids were fixed with 3% glutaraldehyde in a 0.1 M cacodylate buffer, then washed three times with 0.1 M cacodylate buffer, dehydrated using ascending ethanol grade (30, 50, 70, 80, 90, 96, and 100%), and dried in a critical point dryer (CPD 030; BAL-TEC Inc.) using liquid carbon dioxide. The dried organoids were sputtered with gold in a sputter coater (SCD 040; Balzers Union Limited) and observed in a scanning electron microscope (VEGA TS 5136 XM; Tescan Orsay Holding) using a secondary emission detector and a 20-kV acceleration voltage.

## Statistical analysis

All results are expressed as means ± SD of three replicates for each condition. For the testing of normal distribution, Shapiro–Wilk test was used and data were further compared by either paired in case

of RT–qPCR and micro-CT analysis or nonparametric unpaired two-tailed *t* test in case of primary cilia length measurements and Western blot analyses. Differences were considered to be significant at $P < 0.05$.

## Data Availability

Materials described in the article, including all relevant raw data, will be freely available to any researcher wishing to use them for noncommercial purposes, without breaching participant confidentiality.

## Supplementary Information

## Acknowledgements

This work was supported by the Czech Science Foundation (21-05146S) and by the Ministry of Education, Youth and Sports of the Czech Republic (CZ.02.1.01/0.0/0.0/15_003/0000460) to M Buchtova. We acknowledge the core facility CELLIM supported by the Czech-BioImaging large RI project (LM2023050 funded by MEYS CR) for their support with obtaining the scientific data presented in this article. T Barta is supported also by the Ministry of Health (NU22-07-00380).

### Author Contributions

M Dubaic: formal analysis, investigation, visualization, methodology, and writing—original draft.
L Peskova: formal analysis, investigation, visualization, methodology, and writing—original draft.
M Hampl: formal analysis, investigation, methodology, and writing—original draft.
K Weissova: formal analysis, investigation, visualization, and methodology.
C Celiker: formal analysis, investigation, and methodology.
NA Shylo: resources, formal analysis, methodology, and writing—original draft, review, and editing.
E Hruba: formal analysis, investigation, and methodology.
M Kavkova: formal analysis, investigation, methodology, and writing—original draft.
T Zikmund: resources, software, funding acquisition, methodology, and writing—original draft.
SD Weatherbee: resources, methodology, and writing—review and editing.
J Kaiser: resources, software, methodology, and writing—review and editing.
T Barta: conceptualization, data curation, formal analysis, supervision, funding acquisition, investigation, visualization, methodology, and writing—original draft, review, and editing.
M Buchtova: conceptualization, resources, supervision, funding acquisition, methodology, and writing—review and editing.

## Conflict of Interest Statement

The authors declare that they have no conflict of interest.

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
