## [Reviewer comments · Life Science Alliance]

Life Science Alliance

Role of Ciliopathy Protein TMEM107 in Eye Development: Insights from Mouse Model and Retinal Organoid

Marija Dubaic, Lucie Peskova, Marek Hampl, Kamila Weisssova, Canan Celiker, Natalia Shylo, Eva Hrubá, Michaela Kavkova, Tomas Zikmund, Scott Weatherbee, Jozef Kaiser, Tomas Barta, and Marcela Buchtova

DOI: <https://doi.org/10.26508/lsa.202302073>

Corresponding author(s): Marcela Buchtova, Masaryk University and Tomas Barta, Masaryk University

Review Timeline:

Submission Date:	2023-04-04
Editorial Decision:	2023-05-22
Revision Received:	2023-08-20
Editorial Decision:	2023-09-08
Revision Received:	2023-09-27
Accepted:	2023-09-28

Transaction Report:

May 22, 2023

Re: Life Science Alliance manuscript #LSA-2023-02073-T

Marcela Buchtova

Dear Dr. Buchtova,

Thank you for submitting your manuscript entitled "The Role of Ciliopathy Protein TMEM107 in the Eye Development: Insights from Mouse Model and Retinal Organoid" to Life Science Alliance. The manuscript was assessed by expert reviewers, whose comments are appended to this letter. We invite you to submit a revised manuscript addressing the Reviewer comments.

Thank you for this interesting contribution to Life Science Alliance. We are looking forward to receiving your revised manuscript.

Sincerely,

B. MANUSCRIPT ORGANIZATION AND FORMATTING:

Reviewer #1 (Comments to the Authors (Required)):

Dubaic et al in this paper describe the role of TMEM107 in eye development in order to fully understand the function of this gene in which mutations in humans cause ciliopathies. The authors used animal and cellular models lacking TMEM107 expression to complement their study with the aim to precisely define the regulatory role of this gene in the morphogenesis of the eye during development.

This study is really interesting in the field of retinal ciliopathies and well organised, and the conclusions are supported by the results. The authors used a variety of animal and cellular models as well as human retinal organoids, and methods/techniques (CRISPR editing, RNAscope, gene silencing and electron microscopy among others) to shed light more in details in the function of the ciliopathy protein TMEM107 in the eye development. The conclusions are supported by the results described in this manuscript, and these observed results are very useful and important for the field.

There are some comments that the authors need to address:

- The authors need to have a separate section of statistical analysis in the Methods section, where they have to describe in detail how they performed the statistical analysis, the test used, as well as to clarify how they tested the normality and the equality of variances in order to apply the appropriate comparison test.
- In the figure legends the authors need to clearly specify the n number and the statistical test used. In some figures this information is missing (eg Figures 1, 4, 5, 6), so please modify appropriately.
- The authors have generated CRISPR/Cas9 knockout models in hESCs as well as ARPE19 cells (Figures S4 and S5). However, the authors need to check with qPCRs and/or WB that there is no expression of this gene/protein (in the cells and/or ROs), in order to make sure that they have a real KO. Please provide this information.
- In the page 18 (second paragraph) and Figure S7C the authors state and show a 50% downregulation of TMEM107 in the presence of DOX. However, they don't specify if this is mRNA or protein levels. Please provide the information of the technique used (qPCR or WB) as well as the n number and the statistical test used.
- Figure S7E: Please specify the differentiation day of the ROs.
- In the page 18 (end of second paragraph) and Figure S9 the authors talk about upregulation and downregulation. Please provide the quantifications to support these statements.
- In page 21: "While in WT eyes SHH localized to discrete regions in the distal part of the eye (Fig. 7A'), higher expression of this ligand in proximal and distal parts of the NR as well as in optic stalk has been observed (Fig. 7B)." The authors need to add: in the Tmem107^{-/-}.
- There are some english language and typo errors, so please revise the entire text (eg. In page 3 is retinal organoids and not retinoids, page 22 first paragraph etc).
- What kind of mutations have been described in human patients: Missense, Nonsense, Loss of Function, Gain of Function? Could the authors extrapolate their findings with the mutations found in patients?

Reviewer #2 (Comments to the Authors (Required)):

1. Since their discovery, primary cilia organelles have received much attention for their role in cell signaling both during embryonic development as well as in maintaining function of mature cells. It has also become clear that their malfunction can lead to aberrant development and disease. Nevertheless, much remains to be learned about their basic structure and function as well as the unique roles they play in different cell types and pathologies. The present study provides some novel insights into these questions especially with respect to the role of primary cilia in eye development and in particular the role of the transmembrane protein TMEM107 in this process.

2. A major strength of the study is the combination of methods of in vitro and in vivo models including generation of mouse mutants, modified retinal cell lines, and retinal organoid cultures, all of which taken together support a critical role for TMEM107 in ocular development. These results provide insight into human diseases that have been correlated with mutations in this gene and also provide mechanistic insight into the aberrant cellular signaling pathways that may be at fault in these cases. The results will be of general interest to those engaged in study of primary cilia and related human diseases, as well as of specific interest to those who study ocular development and diseases. The data provide strong support for the author's conclusions that TMEM107 is necessary for primary cilia formation and retinal development as mediated through sonic hedgehog signaling pathways.

3. The paper is clearly written and well-documented and supports the main conclusions of the authors.

4. 'Referee Cross-Comments'. Upon consideration of all reviews, there is clear agreement that the work is of interest to the field of retinal development, is generally well-written and the data support the conclusions drawn. However, this reviewer does agree with Reviewer 1 that this manuscript will be further strengthened and more impactful if the queries expressed by Reviewer 1 are addressed by the authors.

Reviewer #1:

Dubaic et al in this paper describe the role of TMEM107 in eye development in order to fully understand the function of this gene in which mutations in humans cause ciliopathies. The authors used animal and cellular models lacking TMEM107 expression to complement their study with the aim to precisely define the regulatory role of this gene in the morphogenesis of the eye during development.

This study is really interesting in the field of retinal ciliopathies and well organised, and the conclusions are supported by the results. The authors used a variety of animal and cellular models as well as human retinal organoids, and methods/techniques (CRISPR editing, RNAscope, gene silencing and electron microscopy among others) to shed light more in details in the function of the ciliopathy protein TMEM107 in the eye development. The conclusions are supported by the results described in this manuscript, and these observed results are very useful and important for the field.

ANSWER: We would like to thank you very much for kind words

There are some comments that the authors need to address:

-The authors need to have a separate section of statistical analysis in the Methods section, where they have to describe in detail how they performed the statistical analysis, the test used, as well as to clarify how they tested the normality and the equality of variances in order to apply the appropriate comparison test.

ANSWER: We added information about statistical analysis into the Methods section.

-In the figure legends the authors need to clearly specify the n number and the statistical test used. In some figures this information is missing (eg Figures 1, 4, 5, 6), so please modify appropriately.

ANSWER: Thank you for pointing this out. The requested information was added to figure legends.

-The authors have generated CRISPR/Cas9 knockout models in hESCs as well as ARPE19 cells (Figures S4 and S5). However, the authors need to check with qPCRs and/or WB that there is no expression of this gene/protein (in the cells and/or ROs), in order to make sure that they have a real KO. Please provide this information.

ANSWER: Detecting TMEM107 using western blot or IHC approaches is challenging due to its low abundance in cells. Therefore, we assessed TMEM107 expression using RT-qPCR. Since mRNA can still be detected even after CRISPR KO (as it continues to be expressed but is likely degraded by nonsense-mediated decay), we designed new RT-qPCR primers. The forward primer was designed to target the CRISPR-edited site in the first exon, while the reverse primer targets the second exon. See new Figure S8 and updated Supplementary Table 1 for primer sequences.

-In the page 18 (second paragraph) and Figure S7C the authors state and show a 50% downregulation of TMEM107 in the presence of DOX. However, they don't specify if this is mRNA or protein levels. Please provide the information of the technique used (qPCR or WB) as well as the n number and the statistical test used.

ANSWER: Thank you for the comment, this escaped our attention. It was mRNA expression, the information was added to the text and figure legend (now Fig. S8)

-Figure S7E: Please specify the differentiation day of the ROs.

ANSWER: Information was added into figure legend (now Fig. S8)

-In the page 18 (end of second paragraph) and Figure S9 the authors talk about upregulation and downregulation. Please provide the quantifications to support these statements.

ANSWER: Quantification analyses were performed and information about statistical analyses included in figure.

-In page 21: "While in WT eyes SHH localized to discrete regions in the distal part of the eye (Fig. 7A'), higher expression of this ligand in proximal and distal parts of the NR as well as in optic stalk has been observed (Fig. 7B)." The authors need to add: in the *Tmem107*^{-/-}.

ANSWER: Information was added.

-There are some english language and typo errors, so please revise the entire text (eg. In page 3 is retinal organoids and not retinoids, page 22 first paragraph etc).

ANSWER: We went through the text carefully and removed all typos.

-What kind of mutations have been described in human patients: Missense, Nonsense, Loss of Function, Gain of Function? Could the authors extrapolate their findings with the mutations found in patients?

ANSWER: Information was added into the text.

.....

Reviewer #2:

1. Since their discovery, primary cilia organelles have received much attention for their role in cell signaling both during embryonic development as well as in maintaining function of mature cells. It has also become clear that their malfunction can lead to aberrant development and disease. Nevertheless, much remains to be learned about their basic structure and function as well as the unique roles they play in different cell types and pathologies. The present study provides some novel insights into these questions especially with respect to the role of primary cilia in eye development and in particular the role of the transmembrane protein TMEM107 in this process.

2. A major strength of the study is the combination of methods of in vitro and in vivo models including generation of mouse mutants, modified retinal cell lines, and retinal organoid cultures, all of which taken together support a critical role for TMEM107 in ocular development. These results provide insight into human diseases that have been correlated with mutations in this gene and also provide mechanistic insight into the aberrant cellular signaling pathways that may be at fault in these cases. The results will be of general interest to those engaged in study of primary cilia and related human diseases, as well as of specific interest to those who study ocular development and diseases. The data provide strong support for the author's conclusions that TMEM107 is necessary for primary cilia formation and retinal development as mediated through sonic hedgehog signaling pathways.

3. The paper is clearly written and well-documented and supports the main conclusions of the authors.

4. 'Referee Cross-Comments'. Upon consideration of all reviews, there is clear agreement that the work is of interest to the field of retinal development, is generally well-written and the data support the conclusions drawn. However, this reviewer does agree with Reviewer 1 that this manuscript will be further strengthened and more impactful if the queries expressed by Reviewer 1 are addressed by the authors.

ANSWER: We would like to thank you very much for kind words about our work, we tried to fulfill all the requirements of Reviewer 1.

September 8, 2023

RE: Life Science Alliance Manuscript #LSA-2023-02073-TR

Dr. Marcela Buchtova
Masaryk University
Kamenice 5
Brno 62500
Czech Republic

Dear Dr. Buchtova,

Thank you for submitting your revised manuscript entitled "Role of Ciliopathy Protein TMEM107 in Eye Development: Insights from Mouse Model and Retinal Organoid". We would be happy to publish your paper in Life Science Alliance pending final revisions necessary to meet our formatting guidelines.

- please note that titles in the system and on the manuscript file must match
- Please upload all figure files as individual ones, including the supplementary figure files; all figure legends should only appear in the main manuscript file
- please add ORCID ID for the secondary corresponding author--they should have received instructions on how to do so
- please consult our manuscript preparation guidelines <https://www.life-science-alliance.org/manuscript-prep> and make sure your manuscript sections are in the correct order
- please incorporate any points from the Conclusion section into the Discussion. We only allow a Discussion section
- please use the [10 author names et al.] format in your references (i.e., limit the author names to the first 10)
- we encourage you to revise the figure legends for figures 3, 5, and 7 such that the figure panels are introduced in alphabetical order
- please add your main, supplementary figure, table, and video legends to the main manuscript text after the references section
- please add callouts for Figures 2C; 3A,C,G,I,K; 5A,D,G,J,M,P; S1A-C; S4; S5A-D; S6A,D; S7A-L to your main manuscript text
- please rename "Availability of data" section to Data Availability

Figure checks:

- in Figure 2, it seems the boxes for the A and F series can be adjusted to appear as better matches for the magnified versions shown in the panels next to them, and panel A" may be inverted

A. FINAL FILES:

-- Summary blurb (enter in submission system): A short text summarizing in a single sentence the study (max. 200 characters including spaces). This text is used in conjunction with the titles of papers, hence should be informative and complementary to the title. It should describe the context and significance of the findings for a general readership; it should be written in the

present tense and refer to the work in the third person. Author names should not be mentioned.

B. MANUSCRIPT ORGANIZATION AND FORMATTING:

Sincerely,

Reviewer #1 (Comments to the Authors (Required)):

The revised manuscript has improved a lot and the authors have addressed all my comments. The manuscript can be accepted for publication.

September 28, 2023

RE: Life Science Alliance Manuscript #LSA-2023-02073-TRR

Dr. Marcela Buchtova
Masaryk University
Kamenice 5
Brno 62500
Czech Republic

Dear Dr. Buchtova,

Thank you for submitting your Research Article entitled "Role of Ciliopathy Protein TMEM107 in Eye Development: Insights from Mouse Model and Retinal Organoid". It is a pleasure to let you know that your manuscript is now accepted for publication in Life Science Alliance. Congratulations on this interesting work.

DISTRIBUTION OF MATERIALS:

Again, congratulations on a very nice paper. I hope you found the review process to be constructive and are pleased with how the manuscript was handled editorially. We look forward to future exciting submissions from your lab.

Sincerely,
